# Observation of emergent momentum–time skyrmions in parity–time-symmetric non-unitary quench dynamics

Kunkun Wang[1,2], Xingze Qiu [3,4], Lei Xiao[1,2], Xiang Zhan[1,2], Zhihao Bian[1,2], Barry C. Sanders [5,6,7], Wei Yi[3,4] & Peng Xue [1,2,8]

Topology in quench dynamics gives rise to intriguing dynamic topological phenomena, which are intimately connected to the topology of static Hamiltonians yet challenging to probe experimentally. Here we theoretically characterize and experimentally detect momentum-time skyrmions in parity-time ($\mathcal{PT}$)-symmetric non-unitary quench dynamics in single-photon discrete-time quantum walks. The emergent skyrmion structures are protected by dynamic Chern numbers defined for the emergent two-dimensional momentum-time sub-manifolds, and are revealed through our experimental scheme enabling the construction of time-dependent non-Hermitian density matrices via direct measurements in position space. Our work experimentally reveals the interplay of $\mathcal{PT}$ symmetry and quench dynamics in inducing emergent topological structures, and highlights the application of discrete-time quantum walks for the study of dynamic topological phenomena.

[1] Beijing Computational Science Research Center, 100084 Beijing, China. [2] Department of Physics, Southeast University, 211189 Nanjing, China. [3] CAS Key Laboratory of Quantum Information, University of Science and Technology of China, 230026 Hefei, China. [4] CAS Center For Excellence in Quantum Information and Quantum Physics, Hefei 230026, China. [5] Institute for Quantum Science and Technology, University of Calgary, Alberta T2N 1N4, Canada. [6] Program in Quantum Information Science, Canadian Institute for Advanced Research, Toronto, Ontario M5G 1Z8, Canada. [7] Shanghai Branch, National Laboratory for Physical Sciences at Microscale, University of Science and Technology of China, 201315 Shanghai, China. [8] State Key Laboratory of Precision Spectroscopy, East China Normal University, 200062 Shanghai, China. Correspondence and requests for materials should be addressed to W.Y. (email: wyiz@ustc.edu.cn) or to P.X. (email: gnep.eux@gmail.com)

Topological phases feature a wealth of fascinating properties governed by the geometry of their ground-state wave functions at equilibrium[1,2], but topological phenomena also manifest as non-equilibrium quantum dynamics in driven-dissipative[3] and Floquet systems[4–7], as well as in quench processes[8–15]. The experimental detection of these dynamic topological phenomena is challenging, since it requires full control and access of the time-evolved state. In recent experiments with ultracold atoms and superconducting qubits, topological objects such as vortices, links, rings, and skyrmions have been identified in the unitary quench dynamics of topological systems via time-resolved and momentum-resolved tomography[16–19]. Here we experimentally establish discrete-time photonic quantum walks (QWs) as another promising arena for engineering and detecting dynamic topological phenomena. Compared to other synthetic systems, the relative ease of introducing loss in photonic systems further enables us to experimentally investigate novel dynamic topological phenomena in the non-unitary regime, where parity–time ($\mathcal{PT}$) symmetry plays an important role.

In discrete-time photonic QWs[20–25], single photons, starting from their initial states, are subject to repeated unitary operations[26]. While QW dynamics support Floquet topological phases (FTPs)[22–25,27,28], discrete-time QWs can also be viewed as a stroboscopic simulation of quench dynamics between FTPs, during which dynamic topological phenomena should occur. As a first endeavor in this direction, two recent experiments report simulation of dynamic quantum phase transitions using QWs[29,30], where non-analyticities in the time evolution of physical observables are associated with changes in dynamic topological order parameters[31]. However, compared to other synthetic systems exhibiting rich dynamic topological structures in higher dimensions[16–19], dynamic topological structures reported in QWs have been limited to one dimension and characterized by topological invariants defined on one-dimensional manifolds.

Here we theoretically characterize and experimentally detect dynamic skyrmion structures, a two-dimensional topological object, in $\mathcal{PT}$-symmetric one-dimensional QWs of single photons. Originally proposed in high-energy physics[32] and later experimentally observed in magnetic and optical configurations[33–35], skyrmions are a type of topologically stable defects featuring a three-component vector field in two dimensions. In QW dynamics, dynamic skyrmions manifest themselves in the momentum–time spin texture of the time-evolved density matrix, and are protected by quantized dynamic Chern numbers in emergent momentum–time submanifolds[12–14]. To detect

dynamic skyrmions, we devise an experimental scheme where time-dependent momentum–space density matrices of spatially non-localized states are constructed based on a combination of interference-based measurements and projective measurements in position space. Such a practice allows for direct measurements of the density matrix at each time step, which significantly reduces the systematic error introduced by the least-square algorithm in tomographic measurements. We confirm the emergence of dynamic skyrmion structures when QW dynamics correspond to quenches between distinct FTPs in the $\mathcal{PT}$-symmetry-unbroken regime, where the dynamics is coherent despite being non-unitary. Effective coherent dynamics is manifested as temporal oscillatory behavior inherent in off-diagonal density-matrix elements. Such oscillatory phenomena reflect the system's ability to fully retrieve information temporarily lost to the environment by $\mathcal{PT}$ dynamics in the unbroken-symmetry regime[36]. By contrast, when the system is quenched into the $\mathcal{PT}$-symmetry-broken regime, skyrmions are absent in the momentum–time space, as the dynamics become incoherent. Our work unveils the fascinating relation between emergent topology and $\mathcal{PT}$-symmetric non-unitary dynamics, and opens up exploration of higher-dimensional dynamic topological structures using QWs.

## Results

**Quench dynamics in $\mathcal{PT}$-symmetric QWs.** We experimentally implement $\mathcal{PT}$-symmetric non-unitary QWs on a one-dimensional lattice $L$ ($L \in \mathbb{Z}$) with single photons in the cascaded interferometric network illustrated in Fig. 1. The corresponding Floquet operator is

$$U = R\left(\frac{\theta_1}{2}\right) S R\left(\frac{\theta_2}{2}\right) M R\left(\frac{\theta_2}{2}\right) S R\left(\frac{\theta_1}{2}\right), \tag{1}$$

where $R(\theta)$ rotates coin states (encoded in the horizontal and vertical polarizations of single photons $|H\rangle$ and $|V\rangle$) by $\theta$ about the $y$-axis, and $S$ moves the photon to neighboring spatial modes depending on its polarization (see "Methods"). The loss operator $M = 1_{\mathrm{w}} \otimes (|+\rangle\langle+| + \sqrt{1-p}|-\rangle\langle-|)$ enforces a partial measurement in the basis $|\pm\rangle = (|H\rangle \pm |V\rangle)/\sqrt{2}$ at each time step with a success probability $p \in [0, 1]$. Here $1_{\mathrm{w}} = \sum_x |x\rangle\langle x|$ with $|x\rangle$ ($x \in L$) denoting the spatial mode. Note that the non-unitary QW driven by $U$ reduces to a unitary one when $p = 0$.

QWs governed by $U$ stroboscopically simulate non-unitary time evolutions driven by the effective Hamiltonian $H_{\mathrm{eff}}$, with $U = \mathrm{e}^{-iH_{\mathrm{eff}}}$. We define the quasienergy $\varepsilon$ and eigenstate $|\psi\rangle$ as

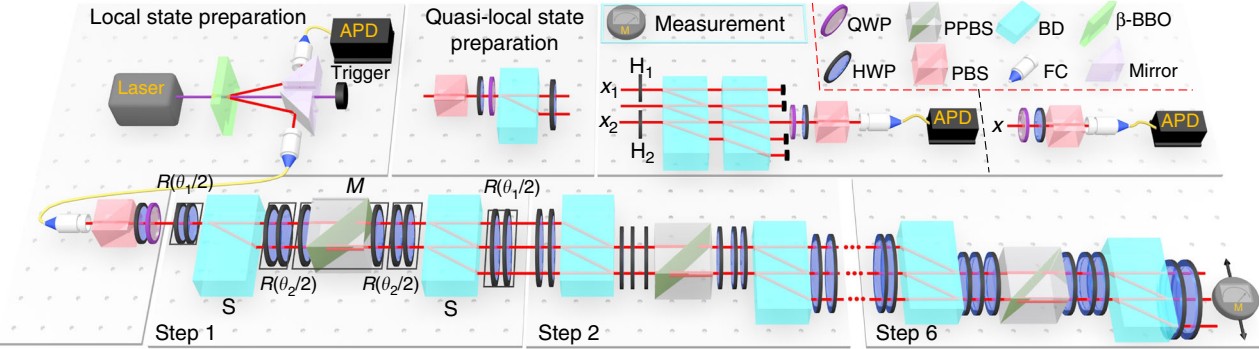

**Fig. 1** Experimental setup for detecting momentum–time skyrmions in non-unitary QWs. Photons are generated via spontaneous parametric down conversion through a Type-I non-linear $\beta$-barium-borate (BBO) crystal. The single signal photon is heralded by the corresponding trigger photon and can be prepared in an arbitrary linear polarization state via a polarizing beam splitter (PBS) and wave plates. Conditional shift operation $S$ and coin rotation $R$ are realized by a beam displacer (BD) and two half-wave plates (HWPs), respectively. For non-unitary QWs, a sandwich-type HWP–PPBS–HWP setup is inserted to introduce non-unitarity, where PPBS is an abbreviation for partially polarizing beam splitters. Two kinds of measurements, including projective measurements and interference-based measurements, are applied before the signal and heralding photons are detected by avalanche photodiodes (APDs)

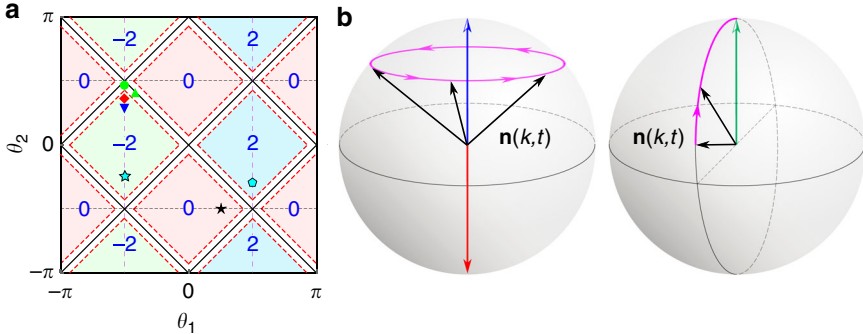

**Fig. 2** Phase diagram and schematic illustrations of non-unitary QW dynamics. **a** Phase diagram for QWs governed by the Floquet operator $U$ in Eq. (1), with the corresponding topological numbers $\nu$ as a function of coin parameters ($\theta_1$, $\theta_2$). Solid black lines are the topological phase boundary, dashed red lines represent boundaries between $\mathcal{PT}$-symmetry-unbroken and $\mathcal{PT}$-symmetry-broken regimes. Black star represents coin parameters of $U^i$, of which the initial state $|\psi^i\rangle$ is an eigenstate. Red diamond and blue triangle correspond to coin parameters of two different final Floquet operators $U^f$ in Fig. 3a, b [Fig. 4a, b], respectively. The cyan filled star and pentagon correspond to coin parameters of $U^i$ and $U^f$ in Fig. 5. The green circle, green triangle correspond to coin parameters of $U^f$ in Fig. 6a, c. **b** Schematic illustrations of the time evolution of $\mathbf{n}(k, t)$ on a Bloch sphere when $E_k^f$ is real (left) and imaginary (right), respectively. Blue and red arrows point to fixed points. The green arrow indicates steady state in the long-time limit. Black arrows indicate the direction of $\mathbf{n}$ ($k$, $t$) at different times

$U|\psi\rangle = \gamma^{-1} e^{-i\varepsilon} |\psi\rangle$, where $\gamma = (1 - p)^{-\frac{1}{4}}$. $U$ possesses passive $\mathcal{PT}$ symmetry with $\mathcal{PT}\gamma U(\mathcal{PT})^{-1} = \gamma^{-1} U^{-1}$, where $\mathcal{PT} = \sum_x |-x\rangle\langle x| \otimes \sigma_3 \mathcal{K}$, $\sigma_3 = |H\rangle\langle H| - |V\rangle\langle V|$, and $\mathcal{K}$ is the complex conjugation. It follows that $\varepsilon$ is entirely real in the $\mathcal{PT}$-symmetry-unbroken regime, and can take imaginary values in regimes when $\mathcal{PT}$ symmetry is spontaneously broken[29,37–39]. $U$ also features topological properties, characterized by winding numbers defined through the global Berry phase[40–42]. We show the topological phase diagram of the system in Fig. 2a, where distinct FTPs are marked by their corresponding winding numbers. The boundaries between $\mathcal{PT}$-symmetry-unbroken and $\mathcal{PT}$-symmetry-broken regimes are also shown in red-dashed lines, with $\mathcal{PT}$-symmetry-broken regimes surrounding topological phase boundaries.

To simulate quench dynamics, we initialize the walker photon in the eigenstate $|\psi^i\rangle$ of a Floquet operator $U^i = e^{-iH^i_{\text{eff}}}$, characterized by coin parameters ($\theta_1^i$, $\theta_2^i$). The walker at the $t$th time step is given by $|\psi(t)\rangle = e^{-iH^f_{\text{eff}}t}|\psi^i\rangle$, such that the resulting QW can be identified as a sudden quench between $H^i_{\text{eff}}$ and $H_{\text{eff}}$. Adopting notations in typical quench dynamics, we denote $U$ and $H_{\text{eff}}$ as $U^f$ and $H^f_{\text{eff}}$ in the following, characterized by coin parameters ($\theta_1^f$, $\theta_2^f$).

**Fixed points and emergent skyrmions.** Due to the lattice translational symmetry of $U^{i,f}$, dynamics in different quasi-momentum $k$-sectors are decoupled. We denote pre-quench and post-quench Floquet operators in each $k$-sector as $U^i_k$ and $U^f_k$, respectively, whose eigenstates are $|\psi^{i,f}_{k,\pm}\rangle$. Quasienergies of $U^{i,f}_k$ are denoted as $\varepsilon^{i,f}_{k,\pm}$, with $\varepsilon^{i,f}_{k,\pm} = \pm E^{i,f}_k$. Focusing on the case where $U^i$ is in the $\mathcal{PT}$-symmetry-unbroken regime, we write the initial state as $|\psi^i_{k,-}\rangle$ in each $k$-sector.

By invoking the biorthogonal basis[43], the non-unitary time evolution of the system is captured by a non-Hermitian density matrix, which can be written as[14]

$$\rho(k,t) = \frac{1}{2}[\tau_0 + \mathbf{n}(k,t)\cdot\boldsymbol{\tau}], \qquad (2)$$

where $\mathbf{n}(k, t) = (n_1, n_2, n_3)$, $\boldsymbol{\tau} = (\tau_1, \tau_2, \tau_3)$, $\tau_i = \sum_{\mu,\nu=\pm} |\psi^f_{k,\mu}\rangle \sigma_i^{\mu\nu} \langle\chi^f_{k,\nu}|$ ($i = 0, 1, 2, 3$), and $\langle\chi^f_{k,\mu}|$ $\left(|\psi^f_{k,\mu}\rangle\right)$ is the left (right) eigenvector of $U^f_k$. Here $\sigma_0$ is a $2 \times 2$ identity matrix, and $\sigma_i$ ($i = 1, 2, 3$) is the corresponding standard Pauli matrix.

A key advantage of adopting Eq. (2) is that $\mathbf{n}(k, t)$ becomes a real unit vector, which enables us to visualize the non-unitary dynamics on a Bloch sphere. As illustrated in Fig. 2b, when $E_k^f$ is real, $\mathbf{n}(k, t)$ rotates around poles of the Bloch sphere with a period $t_0 = \pi/E_k^f$. Thus, momenta corresponding to poles of the Bloch sphere are identified as two different kinds of fixed points, where the density matrices do not evolve in time. In contrast, when $E_k^f$ is imaginary, there are no fixed points in the dynamics, as $\mathbf{n}(k, t)$ asymptotically approaches the north pole in the long-time limit (see Fig. 2b).

When $U^i$ and $U^f$ belong with distinct FTPs in the $\mathcal{PT}$-symmetry-unbroken regime, fixed points of different kinds necessarily emerge in pairs[14,29]. Each momentum submanifold between a pair of distinct fixed points can be combined with the $S^1$ topology of the periodic time evolution to form an emergent $S^2$ momentum–time manifold, which can be mapped to the $S^2$ Bloch sphere of $\mathbf{n}(k, t)$. The Chern number characterizing such an $S^2 \to S^2$ mapping is finite and gives rise to intriguing skyrmion structures in the emergent momentum–time manifolds.

To probe fixed points and momentum–time skyrmions, we perform projective measurements and interference-based measurements to construct the Hermitian density matrix $\rho'(k, t) = |\psi_k(t)\rangle\langle\psi_k(t)|$. This is achieved by writing $\rho'(k,t) = \frac{1}{2}\sum_{j=0}^{3}\sum_{x_1,x_2} e^{-ik(x_1-x_2)}\langle\psi_{x_2}(t)|\sigma_j|\psi_{x_1}(t)\rangle\sigma_j$, where $|\psi_x(t)\rangle$ is the coin state on site $x$ at the $t$th time step. We experimentally measure $\langle\psi_{x_2}(t)|\sigma_j|\psi_{x_1}(t)\rangle$ ($j = 0, 1, 2, 3$) for each pair of positions $x_1$ and $x_2$ directly. We then calculate the non-Hermitian density matrix $\rho(k, t)$ from $\rho'(k, t)$ and determine $\mathbf{n}(k, t)$ through $\mathbf{n}(k, t) = \text{Tr}[\rho(k, t)\boldsymbol{\tau}]$.

Whereas the Hermitian density matrix $\rho'(k, t)$ is experimentally accessible, visualization of non-unitary dynamics on the Bloch sphere starting from $\rho'(k, t)$ requires normalization. It is also difficult to characterize the connection between fixed points and dynamic skyrmions using $\rho'(k, t)$. By contrast, such a connection is revealed elegantly with the biorthogonal construction of $\rho(k, t)$, as we detail in "Methods" and Supplementary Note 4. This is why we characterize skyrmions using the spin texture $\mathbf{n}(k, t)$ associated with $\rho(k, t)$.

**Dynamics in the $\mathcal{PT}$-symmetry-unbroken regime.** We first study fixed points and momentum–time skyrmions in the

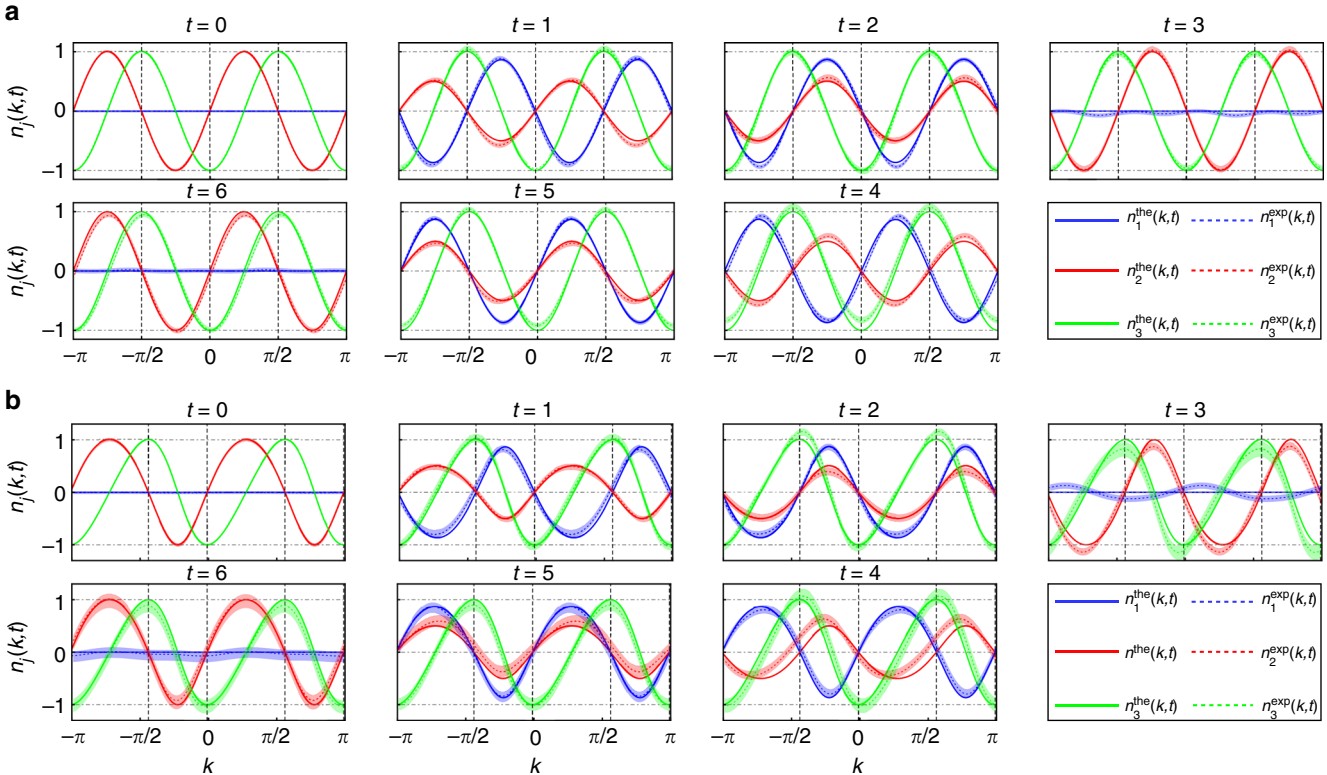

**Fig. 3** Experimental results of $\mathbf{n}(k, t)$. Time-evolution of $\mathbf{n}(k, t)$ up to $t = 6$ for quench processes between (**a**) an initial unitary Floquet operator characterized by $(\theta_1^i = \pi/4, \theta_2^i = -\pi/2)$ and a final unitary Floquet operator characterized by $(\theta_1^f = -\pi/2, \theta_2^f = \pi/3)$ and (**b**) an initial non-unitary Floquet operator characterized by $(\theta_1^i = \pi/4, \theta_2^i = -\pi/2)$ and a final non-unitary Floquet operator characterized by $(\theta_1^f = -\pi/2, \theta_2^f = \arcsin(\frac{1}{\alpha}\cos\frac{\pi}{6}))$. The period of oscillations is $t_0 = 6$ for all $k$. Fixed points (vertical dashed lines) are located at $\{-\pi, -\pi/2, 0, \pi/2\}$ for unitary dynamics (**a**) and at $\{-0.4399\pi, -0.0099\pi, 0.5601\pi, 0.9901\pi\}$ for non-unitary dynamics (**b**). Shading indicates experimental error bars that are due to photon-counting statistics

$\mathcal{PT}$-symmetry-unbroken regime. For comparison, we also experimentally characterize these quantities in unitary dynamics. We initialize the walker on a localized lattice site $|x = 0\rangle$ and in the coin state $|\psi_-^i\rangle_c$. Here, $|x\rangle$ denotes the spatial mode. Importantly, $\left|\psi_{k,-}^i\right\rangle = \left|\psi_-^i\right\rangle_c$ is an eigenstate of $U_k^i$ for all $k$, with the corresponding $(\theta_1^i, \theta_2^i)$ on black dashed lines in Fig. 2a. Without loss of generality, we choose $(\theta_1^i = \pi/4, \theta_2^i = -\pi/2)$ for both the unitary and non-unitary cases.

For the first case of study, we implement unitary QWs with $|\psi_-^i\rangle_c = (|H\rangle + i|V\rangle)/\sqrt{2}$ and $(\theta_1^f = -\pi/2, \theta_2^f = \pi/3)$, which simulate quench processes between FTPs with $\nu^i = 0$ and $\nu^f = -2$. We have chosen $(\theta_1^f, \theta_2^f)$ on purple dashed lines, where qusienergy bands are flat. Oscillatory dynamics of $\mathbf{n}(k, t)$ in different $k$-sectors thus feature the same period, as illustrated in Fig. 3a. We identify fixed points of unitary dynamics at high-symmetry points of the Brillioun zone $\{-\pi, -\pi/2, 0, \pi/2\}$, where $\mathbf{n}(k, t)$ become independent of time.

For the second case of study, we implement non-unitary QWs with $p = 0.36$, $|\psi_-^i\rangle_c = 0.7606|H\rangle + 0.6492i|V\rangle$, and $[\theta_1^f = -\pi/2, \theta_2^f = \arcsin(\frac{1}{\alpha}\cos\frac{\pi}{6})]$ (here $\alpha = \frac{\gamma}{2}(1 + \sqrt{1-p})$). The post-quench FTP is in the $\mathcal{PT}$-symmetry unbroken regime with $\nu^f = -2$. As shown in Fig. 3b, dynamics of $\mathbf{n}(k, t)$ is still oscillatory, but fixed points are shifted away from the high-symmetry points, consistent with theoretical predictions.

In Fig. 4, we plot $\mathbf{n}(k, t)$ in the momentum–time space. The oscillatory behavior in $\mathbf{n}(k, t)$ is then manifested as momentum–time skyrmions, which are protected by dynamic Chern numbers defined on the corresponding momentum–time submanifold. By contrast, when the system

is quenched between FTPs with the same winding number, skyrmion-lattice structures are no longer present, as shown in the Supplementary Fig. 4. Comparing Fig. 4a, b, we see that dynamic skyrmion structures in the unitary and the $\mathcal{PT}$-symmetric non-unitary quench processes are qualitatively similar; albeit, in the non-unitary case, skyrmion structures are slightly deformed due to the shift of fixed points. However, as we show later, skyrmion structures are generically absent when the system is quenched into the $\mathcal{PT}$-symmetry-broken regime.

Our configuration is also capable of simulating quench dynamics between topologically non-trivial FTPs. To demonstrate this, we implement non-unitary QWs with $|\psi_-^i\rangle = (-0.1111 - 0.6983i)|0\rangle \otimes |+\rangle - \frac{1}{\sqrt{2}}|-2\rangle \otimes |-\rangle$ and $(\theta_1^f = \pi/2, \theta_2^f = -\arcsin(\frac{1}{\alpha}\cos\frac{\pi}{6}))$. The initial state is the eigenstate of $U^i$ with $(\theta_1^i = -\pi/2, \theta_2^i = -\pi/4)$, which is prepared by performing gate operations prior to QW dynamics (see "Methods"). The pre-quench and post-quench FTPs are in the $\mathcal{PT}$-symmetry unbroken regime with $\nu^i = -2$ and $\nu^f = 2$, respectively. In Fig. 5, we show the corresponding oscillations of $\mathbf{n}(k, t)$, as well as the momentum–time skyrmions. Consistent with theoretical predictions (see "Methods"), a total of eight fixed points exist in the dynamics, which divide the first Brillioun zone into eight submanifolds with skyrmions emerging in momentum–time space in between each adjacent pair of fixed points.

**Dynamics in the $\mathcal{PT}$-symmetry-broken regime.** We now turn to the case where $U^f$ belong with the $\mathcal{PT}$-symmetry-broken regime. We first initialize the walker on a localized lattice site in the coin

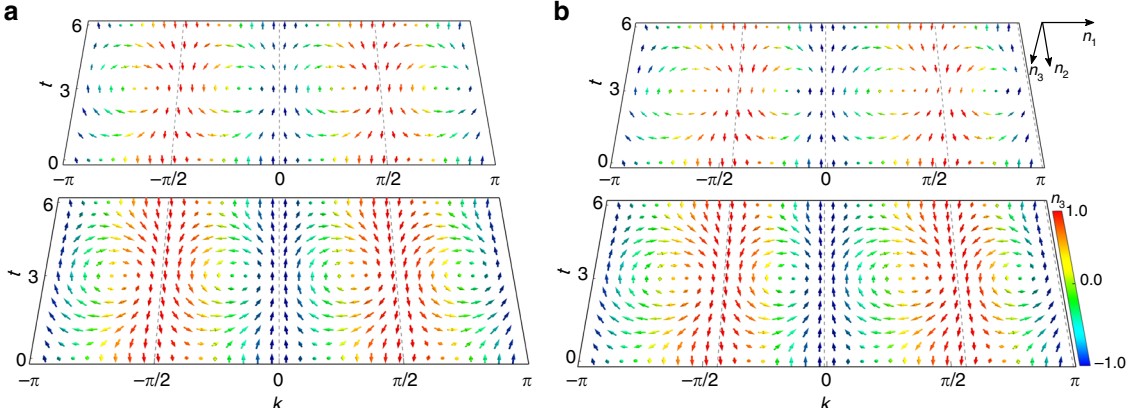

**Fig. 4** Experimental results of spin texture $\mathbf{n}(k, t)$. Experimental (upper layer) and theoretical results (lower layer) of spin texture $\mathbf{n}(k, t)$ in the momentum–time space for quench processes corresponding to (**a**) Fig. 3a, and (**b**) Fig. 3b, respectively. The temporal resolution of experimental measurements is limited by discrete time steps of QWs, whereas we apply a better resolution in theoretical results for a clearer view of skyrmions. The spin textures are colored according to $n_3(k, t)$ and fixed points are indicated by dashed lines

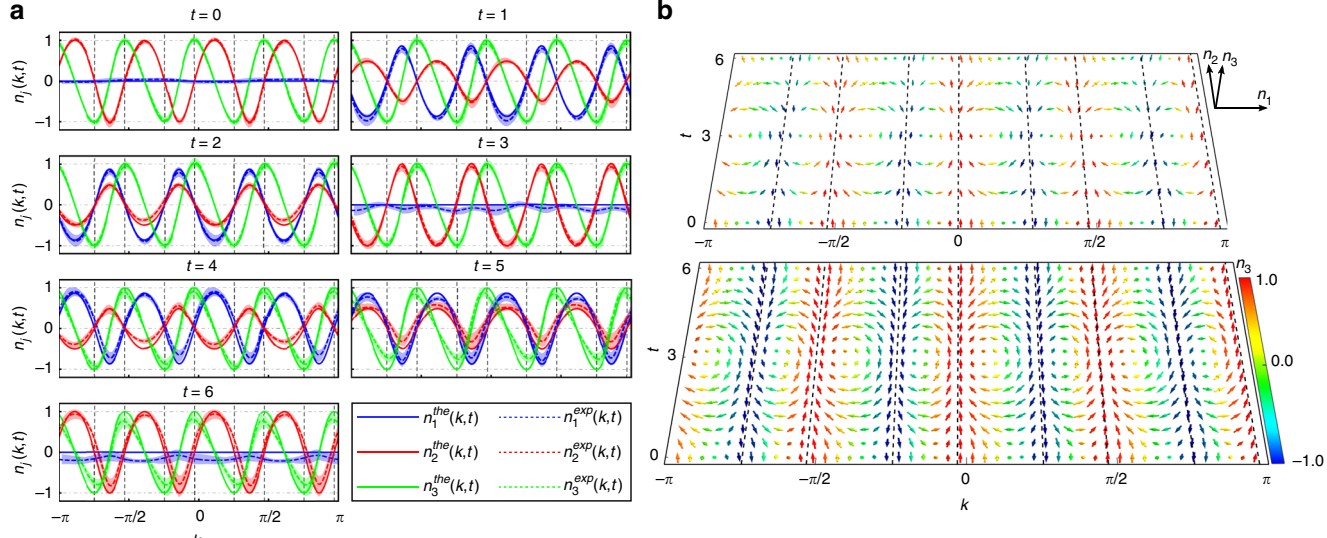

**Fig. 5** Experimental results of the $\mathcal{PT}$-symmetry-unbroken QW dynamics with a large winding-number difference. **a** Time-evolution and **b** spin textures of $\mathbf{n}(k, t)$ in momentum–time space for a quench process between the initial non-unitary Floquet operator given by $(\theta_1^i = -\pi/2, \theta_2^i = -\pi/4)$ with $\nu^i = -2$, and the final one characterized by $(\theta_1^f = \pi/2, \theta_2^f = -\arcsin(\frac{1}{\alpha}\cos\frac{\pi}{6}))$ with $\nu^f = 2$. The eight fixed points (dashed lines) are located at $\{-0.5300\pi, -0.0300\pi, 0.4699\pi, 0.9699\pi\}$ for $c_- = 0$; and $\{-0.7450\pi, -0.2450\pi, 0.2550\pi, 0.7550\pi\}$ for $c_+ = 0$. The color code represents the value of $n_3(k, t)$

state $(|H\rangle + |V\rangle)/\sqrt{2}$ and evolve it under $U^f$ characterized by $(\theta_1^f = -\pi/2, \theta_2^f = \frac{1}{2}(\pi - \arccos\frac{1}{\alpha}))$, which is in the $\mathcal{PT}$-symmetry-broken regime with a completely imaginary quasienergy spectrum. As shown in Fig. 6a, there is no periodical evolution in $\mathbf{n}(k, t)$ anymore. Instead, different components of $\mathbf{n}(k, t)$ slowly approach a steady state with $\mathbf{n} = (0, 0, 1)$ in the long-time limit. This is more clearly seen in momentum–time space shown in Fig. 6b, where skyrmion structures are absent and vectors in all $k$-sectors tend to point out of the plane in the long-time limit. We note that dynamics of $\mathbf{n}(k, t)$ here is insensitive to the choice of initial state, as the system always relaxes to the steady state at long times.

We then start from the same initial state as in Fig. 3b and evolve it under a different $U^f$ characterized by $(\theta_1^f = -0.39\pi, \theta_2^f = 0.3864\pi)$, which is in the $\mathcal{PT}$-symmetry-broken regime with a partially real (and partially imaginary) quasienergy spectrum. As shown in Fig. 6c, d, whereas the evolution of $\mathbf{n}(k, t)$ is still periodic for $k$-sectors with real quasienergies, the existence of steady-state approaching $k$-sectors

with imaginary quasienergies deform spin textures in those sectors and destroy the overall dynamic skyrmion structures in the momentum–time space.

## Discussion

By simulating quench dynamics of topological systems using photonic QWs, we have revealed emergent momentum–time skyrmions, protected by dynamic Chern numbers defined on the momentum–time submanifolds. These emergent topological phenomena are underpinned by fixed points of dynamics, which can exist for both unitary and non-unitary quench processes if the system is quenched across a topological phase boundary. We have further confirmed the decisive role of $\mathcal{PT}$-symmetry on the existence of fixed points and skyrmions in non-unitary dynamics.

Emergent momentum–time skyrmions and dynamic Chern numbers reported here are intimately connected with topological entanglement-spectrum crossing[13], as well as the recently observed dynamic quantum phase transitions and dynamic topological order parameters[19,29,30]. In particular, whereas

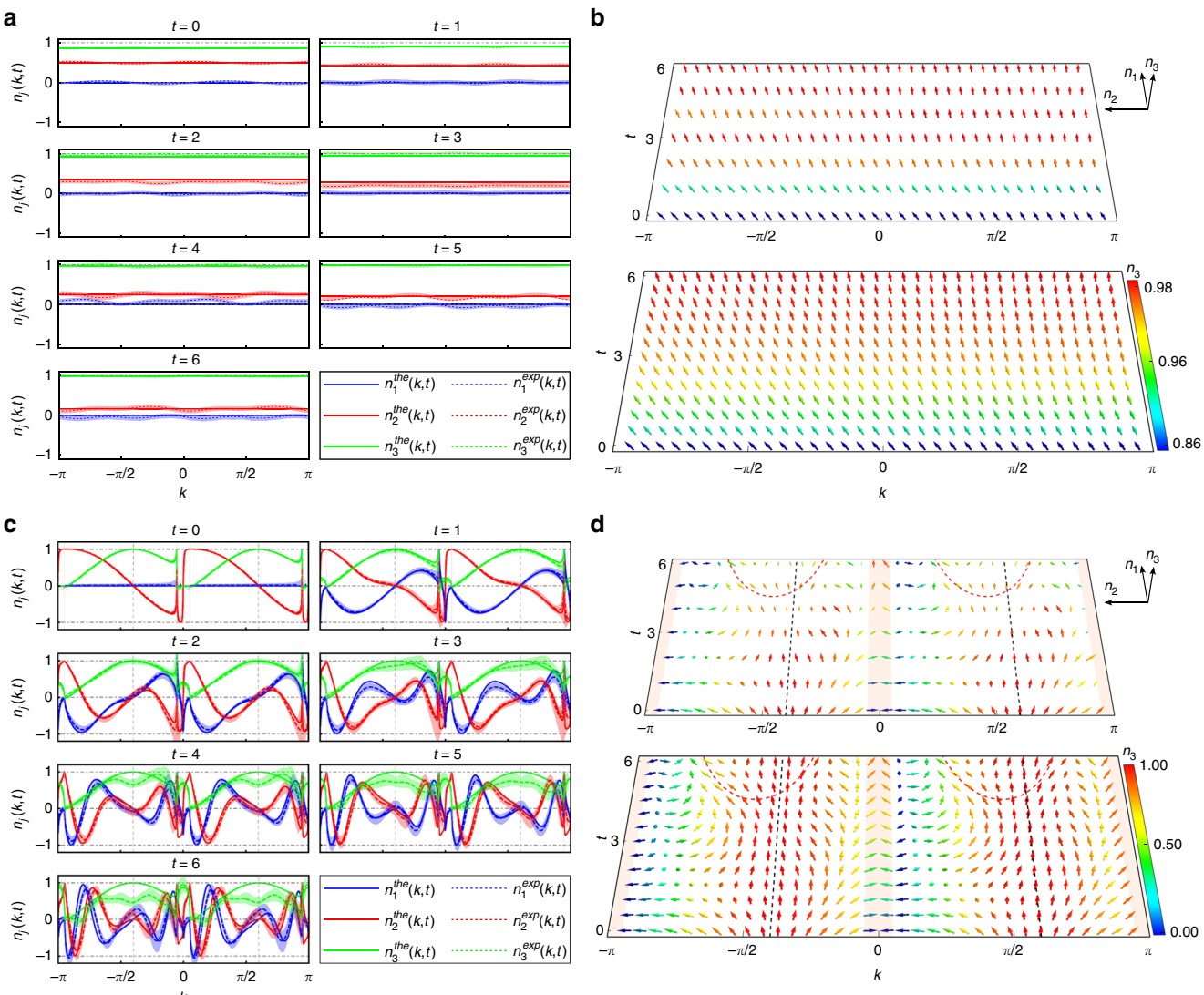

**Fig. 6** Experimental results for the $\mathcal{PT}$-symmetric broken QW dynamics. **a** Time-evolution and (**b**) spin textures of $\mathbf{n}(k, t)$ in momentum–time space for a quench process between the initial non-unitary Floquet operator given by $(\theta_1^i = \pi/4, \theta_2^i = -\pi/2)$ and the final $\mathcal{PT}$-symmetry-broken Floquet operator given by $\left[\theta_1^f = -\pi/2, \theta_2^f = \frac{1}{2}\left(\pi - \arccos\frac{1}{a}\right)\right]$. The quasi-energy spectrum associated with $U^f$ is completely imaginary in this case. (**c**) Time-evolution and (**d**) spin textures of $\mathbf{n}(k, t)$ in the momentum–time space when the system is quenched from the same initial state as in Fig. 3b into a final $\mathcal{PT}$-symmetry-broken Floquet operator given by $(\theta_1^f = -0.39\pi, \theta_2^f = 0.3864\pi)$ with $\nu^f = -2$. The quasi-energy spectrum associated with $U^f$ features purely imaginary [red shaded areas in (**d**)] and purely real regions in momentum space. Only two fixed points of the same kind exist at $\{-0.4000\pi, 0.6000\pi\}$ (dashed lines), whereas no skyrmion structures exist in any momentum–time submanifold. The spin vectors in (**c**) and (**d**) are colored according to $n_3(k, t)$

dynamic topological order parameters and dynamic Chern numbers originate from completely different topological constructions, both are hinged upon the presence of fixed points in quench dynamics[14,19]. With the highly flexible control of photonic QW protocols, it would be interesting to investigate dynamic topological phenomena in higher dimensions or associated with other topological classifications in the future[44]. Our work thus paves the way for a systematic experimental study of dynamic topological phenomena in both unitary and non-unitary dynamics.

## Methods

**Experimental implementation of $U$.** We implement the coin operator $R(\theta) = 1_w \otimes e^{-i\theta\sigma_2}$, the shift operator $S = \sum_x(|x-1\rangle\langle x| \otimes |H\rangle\langle H| + |x+1\rangle \langle x| \otimes |V\rangle\langle V|)$, and the loss operator $M = 1_w \otimes (|+\rangle\langle+| + \sqrt{1-p}|-\rangle\langle-|)$, following the approach outlined in ref.[29]. Here, $|\pm\rangle = (|H\rangle \pm |V\rangle)/\sqrt{2}$, $\sigma_2 = i(-|H\rangle \langle V| + |V\rangle\langle H|)$ is the standard Pauli operator under the polarization basis, $x$ ($x \in L$)

denotes the spatial mode, $1_w = \sum_x |x\rangle\langle x|$, and the loss parameter $p = 0.36$ for non-unitary QWs in our experiment.

**Preparation of quasi-local initial state.** To simulate quench dynamics between topologically non-trivial FTPs in Fig. 5, the QW dynamics starts from a quasi-local initial state $|\psi_-^i\rangle = (-0.1111 - 0.6983i)|0\rangle \otimes |+\rangle - \frac{1}{\sqrt{2}}|-2\rangle \otimes |-\rangle$, which is a $U^i$ eigenstate with $\nu^i = -2$. As illustrated in Fig. 1, the state is experimentally prepared in the following way. First, starting from $|x = 0\rangle$, the polarization of single photons is prepared in $\left(\frac{1}{\sqrt{2}}, -0.1111 - 0.6983i\right)^T$ by tuning the setting angles of the wave plates right after the first PBS. Then we use a BD to split the photons into the spatial modes $|x = 0\rangle$ and $|x = -2\rangle$ depending on their polarizations. Finally, a HWP at a fixed angle rotates polarizations of single photons in both spatial modes.

**$\mathcal{PT}$-symmetric non-unitary QW.** The non-unitary Floquet operator $U$ in Eq. (1) has passive $\mathcal{PT}$ symmetry, from which we can define $\tilde{U} = \gamma U$ with $\gamma = (1-p)^{-\frac{1}{4}}$. $\tilde{U}$ has active $\mathcal{PT}$ symmetry, with the symmetry operator $\mathcal{PT} = \sum_x |-x\rangle \langle x| \otimes \sigma_3\mathcal{K}$, where $\mathcal{K}$ is the complex conjugation. As homogeneous QWs have lattice

translational symmetry, we write $\tilde{U}$ in momentum space

$$\tilde{U}_k = d_0 \sigma_0 - id_1\sigma_1 - id_2\sigma_2 - id_3\sigma_3,$$
$$d_0 = \alpha[\cos(2k)\cos\theta_1\cos\theta_2 - \sin\theta_1\sin\theta_2],$$
$$d_1 = i\beta, \tag{3}$$
$$d_2 = \alpha[\cos(2k)\cos\theta_2\sin\theta_1 + \cos\theta_1\sin\theta_2],$$
$$d_3 = -\alpha\sin(2k)\cos\theta_2,$$

where $\alpha = \frac{\gamma}{2}(1+\sqrt{1-p}), \beta = \frac{\gamma}{2}(1-\sqrt{1-p})$.

Eigenvalues of $\tilde{U}_k$ are $\lambda_{k,\pm} = d_0 \mp i\sqrt{1-d_0^2}$, and the corresponding quasienergy $\varepsilon_{k,\pm} = i\ln(\lambda_{k,\pm})$. When $d_0^2 < 1$ for all $k$, the quasienergy is real, and the system is in the $\mathcal{PT}$-symmetry-unbroken regime. Whereas if $d_0^2 \geq 1$ for some $k$, the $\mathcal{PT}$ symmetry is spontaneously broken and the quasienergy is imaginary in the corresponding momentum range.

**Winding numbers of non-unitary QWs.** Non-unitary QWs governed by $U$ possess topological properties, which are characterized by winding numbers defined through the global Berry phase $\nu = \varphi_B/2\pi$. Here $\varphi_B = \varphi_{Z+} + \varphi_{Z-}$, with generalized Zak phases

$$\varphi_{Z\pm} = -i\oint dk \frac{\langle \chi_{k,\pm} | \frac{d}{dk} | \psi_{k,\pm}\rangle}{\langle \chi_{k,\pm} | \psi_{k,\pm}\rangle}. \tag{4}$$

The integral above is over the first Brillouin zone and $\langle \chi_{k,\mu}|$ and $|\psi_{k,\mu}\rangle$ ($\mu = \pm$) are, respectively, the left and right eigenstates of $U_k$, defined through $U_k^\dagger|\chi_{k,\mu}\rangle = \lambda_\mu^*|\chi_{k,\mu}\rangle$ and $U_k|\psi_{k,\mu}\rangle = \lambda_\mu|\psi_{k,\mu}\rangle$), respectively.

Besides $\mathcal{PT}$ symmetry, the system also possesses particle–hole symmetry $\mathcal{K}U\mathcal{K}^{-1} = U$. According to refs. [27,45], the complete topological characterization of the system should be $Z \oplus Z$, where two topological invariants exist. Following ref. [46], we define another winding number $\nu'$ by choosing a different time frame with Floquet operator $U' = M^{\frac{1}{2}}R(\frac{\theta_2}{2})SR(\theta_1)SR(\frac{\theta_2}{2})M^{\frac{1}{2}}$, where $M^{\frac{1}{2}} = 1_w \otimes \left(|+\rangle\langle+| + (1-p)^{\frac{1}{4}}|-\rangle\langle-|\right)$. Note that $\nu'$ can be calculated from the global Berry phase associated with $U'$. Combinations of $\nu$ and $\nu'$ then give the bulk topological invariants $(\nu_0, \nu_\pi) = [(\nu+\nu')/2, (\nu-\nu')/2]$[46], which dictate the correct number of edge states with quasienergies $\text{Re}\,\varepsilon = 0$ and $\text{Re}\,\varepsilon = \pi$, respectively, through the bulk-boundary correspondence. However, for the quench dynamics of homogeneous QWs with no boundaries, we find that it is sufficient to discuss the interplay between existence of momentum–time skyrmions, the pre-quench and post-quench winding numbers, and $\mathcal{PT}$ symmetry in a single time frame. Indeed, as we demonstrate experimentally in Supplementary Note 2, whereas spin textures and locations of fixed points are quantitatively different under different time frames, going to a different time frame does not qualitatively change the conclusions in the main text, as long as the appropriate winding number is used.

**Fixed points and dynamical Chern numbers in QW dynamics.** Non-unitary time evolution of the system is captured by the non-Hermitian density matrix

$$\rho(k,t) := \frac{|\psi_k(t)\rangle\langle\chi_k(t)|}{\langle\chi_k(t)|\psi_k(t)\rangle} = \frac{1}{2}[\tau_0 + \mathbf{n}(k,t)\cdot\boldsymbol{\tau}], \tag{5}$$

where the time-evolved state $|\psi_k(t)\rangle = \sum_{\mu=\pm} c_\mu e^{-i\varepsilon_{k,\mu}^f t}|\psi_{k,\mu}^f\rangle$, the associated state $\langle\chi_k(t)| := \sum_\mu c_\mu^* e^{i\varepsilon_{k,\mu}^{f*}t}\langle\chi_{k,\mu}^f|$, $c_\mu = \langle\chi_{k,\mu}^f|\psi_{k,-}^i\rangle$, and $\langle\chi_{k,\mu}^f| \left(|\psi_{k,\mu}^f\rangle\right)$ is the left (right) eigenvector of $U_k^f$, with the biorthonormal conditions $\langle\chi_{k,\mu}^f|\psi_{k,\nu}^f\rangle = \delta_{\mu\nu}$ and $\sum_\mu|\psi_{k,\mu}^f\rangle\langle\chi_{k,\mu}^f| = 1$. It follows that $\mathbf{n}(k,t) = \text{Tr}[\rho(k,t)\,\boldsymbol{\tau}]$, with $\{\tau_i\}$ satisfying the standard $\mathfrak{su}(2)$ commutation relations.

Following the convention of the main text, we denote the final Flouqet operator in each quasimomentum $k$-sector as $U_k^f$ and the corresponding quasienergy as $\pm E_k^f$. When $E_k^f$ is real, we have

$$n_0 = c_+^* c_+ + c_-^* c_-,$$
$$n_1 = \frac{1}{n_0}(c_+^* c_+ e^{-i2E_k^f t} + \text{c.c.}),$$
$$n_2 = \frac{i}{n_0}(c_+^* c_+ e^{-i2E_k^f t} - \text{c.c.}), \tag{6}$$
$$n_3 = \frac{1}{n_0}(c_+^* c_+ - c_-^* c_-).$$

By contrast, when $E_k^f$ is imaginary, and assuming $\text{Im}(E_k^f) > 0$, we have

$$n_0 = c_+^* c_+ e^{-i2E_k^f t} + c_-^* c_- e^{i2E_k^f t},$$
$$n_1 = \frac{1}{n_0}(c_+^* c_+ + \text{c.c.}),$$
$$n_2 = \frac{i}{n_0}(c_+^* c_+ - \text{c.c.}), \tag{7}$$
$$n_3 = \frac{1}{n_0}(c_+^* c_+ e^{-i2E_k^f t} - c_-^* c_- e^{i2E_k^f t}).$$

From these expressions, it is straightforward to visualize dynamics of $\mathbf{n}(k,t)$ on a Bloch sphere as illustrated in Fig. 2b and discussed in the main text. In particular, when $U^f$ is in the $\mathcal{PT}$-symmetry-unbroken regime, fixed points occur at momenta

with $c_- = 0$ or $c_+ = 0$, which we identify as two different types of fixed points. The total number of these fixed points with $c_+ = 0$ or $c_- = 0$ which are topologically protected is $|\nu^f - \nu^i|$ each[14,29]. As a concrete example, for the quench illustrated in Fig. 5 with $\nu^i = -2$ and $\nu^f = 2$, we observe eight topologically protected fixed points, with half of them $c_- = 0$ and the other half $c_+ = 0$. It follows that dynamic skyrmions exist in momentum–time space in between each adjacent pair of fixed points of different kinds.

**Dynamic Chern number.** When $U_f$ is in the $\mathcal{PT}$-symmetry-unbroken regime, periodic evolution of the density matrix in each $k$-sector gives rise to a temporal $S^1$ topology. In the presence of fixed points, each momentum submanifold between two adjacent fixed points can be combined with the $S^1$ topology in time to form a momentum–time submanifold $S^2$, which can be mapped to the Bloch sphere associated with the vector $\mathbf{n}(k,t)$. These $S^2 \to S^2$ mappings define a series of dynamic Chern numbers

$$C_{mn} = \frac{1}{4\pi}\int_{k_m}^{k_n}dk\int_0^{t_0}dt[\mathbf{n}(k,t)\times\partial_t\mathbf{n}(k,t)]\cdot\partial_k\mathbf{n}(k,t), \tag{8}$$

where $k_m$ and $k_n$ denote two neighboring fixed points, and $t_0 = \pi/E_k^f$. For quenches between Hamiltonians with different winding numbers, dynamic Chern numbers are quantized, with values dependent on the nature of fixed points at $k_m$ and $k_n$: $C_{mn} = 1$ when $c_+(k_m) = 0$ and $c_-(k_n) = 0$; $C_{mn} = -1$ when $c_-(k_m) = 0$ and $c_+(k_n) = 0$. When the two fixed points are of the same kind, $C_{mn} = 0$.

According to its definition in Eq. (8), a finite Chern number in a momentum–time submanifold corresponds to the existence of momentum–time skyrmions in the same submanifold.

**Constructing density matrix from direct measurements.** The non-Hermitian density matrix $\rho(k,t)$ is related to the Hermitian one $\rho'(k,t) := |\psi_k(t)\rangle\langle\psi_k(t)|$ through

$$\rho(k,t) = \frac{\rho'(k,t)\sum_{\mu=\pm}|\chi_{k,\mu}^f\rangle\langle\chi_{k,\mu}^f|}{\text{Tr}[\rho'(k,t)\sum_{\mu=\pm}|\chi_{k,\mu}^f\rangle\langle\chi_{k,\mu}^f|]}, \tag{9}$$

where we have used the biorthonormal conditions $\langle\chi_{k,\mu}^f|\psi_{k,\nu}^f\rangle = \delta_{\mu\nu}$ and $\sum_{\mu=\pm}|\psi_{k,\mu}^f\rangle\langle\chi_{k,\mu}^f| = 1$.

As we have discussed in the main text, we first determine $\rho'(k,t)$ by measuring $\langle\psi_{x_2}(t)|\sigma_j|\psi_{x_1}(t)\rangle$ ($j = 0, 1, 2, 3$) for each pair of positions $x_1$ and $x_2$. In the case of $x_1 = x_2$, we perform projective measurements on the polarizations of photons at each position. In the case of $x_1 \neq x_2$, we employ interference-based measurements to construct the matrix element $\langle\psi_{x_2}(t)|\sigma_j|\psi_{x_1}(t)\rangle$ from experimental data. We then construct $\rho(k,t)$ and $\mathbf{n}(k,t)$ from $\rho'(k,t)$ using Eq. (9).

We note that dynamic skyrmions are also visible in spin textures defined through $\mathbf{n}'(k,t) = \text{Tr}[\rho'(k,t)]$ after enforcing normalization on $\mathbf{n}'(k,t)$. Here the Pauli vector $\boldsymbol{\sigma} = (\sigma_1, \sigma_2, \sigma_3)$. As we demonstrate explicitly in Supplementary Fig. 5, whereas $\mathbf{n}(k,t)$ and $\mathbf{n}'(k,t)$ are distinct from each other, dynamic topological structures inherent in these two spin textures are equivalent and emerge under the same conditions.

## Data availability

Experimental data, any related experimental background information not mentioned in the text and other findings of this study are available from the corresponding author upon reasonable request.

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

## Acknowledgements

This work has been supported by the Natural Science Foundation of China (Grant Nos. 11674056 and 11522545) and the Natural Science Foundation of Jiangsu Province (Grant No. BK20160024). W.Y. acknowledges support from the National Key Research and Development Program of China (Grant Nos. 2016YFA0301700 and 2017YFA0304100).

## Author contributions

K.K.W. performed the experiments with contributions from L.X., X.Z. and Z.H.B. W.Y. developed the theoretical aspects and performed the theoretical analysis with contribution from X.Z.Q. and wrote part of the paper. P.X. designed the experiments, analyzed the results, and wrote part of the paper. B.C.S. revised the paper. K.K.W. and X.Z.Q. contributed equally to this work.

## Additional information

**Competing interests:** The authors declare no competing interests.

