## [Peer Review File · Nature Communications]

Reviewers' comments:

Reviewer #1 (Remarks to the Author):

The paper presents original and interesting work. My only suggestion to the authors is to make the manuscript more readable. In the current version of the paper presentation is too technical.

Reviewer #2 (Remarks to the Author):

The authors report an experiment on observing momentum-time skyrmions in parity-symmetric non-unitary photonic quantum walks. This experiment basically demonstrates the theoretical predictions in Refs. [12-14], which unveil the dynamic Chern number and momentum-time skyrmions underlying the nonequilibrium dynamics of one-dimensional free fermions quenched across different symmetry-protected topological phases. Unlike the continuous quench dynamics considered in Refs. [12-14], the authors deal with a unitary U with fine-tuned parameters such that U^6 is (proportional to) the identity. This unitary thus stroboscopically mimics a flat-band quench dynamics. In the presence of non-unitarity, which is controlled by a HWP-PPBS-HWP setup, almost the same skyrmion spin textures $n(k,t)$ as the unitary case are observed in the PT -symmetry-unbroken regime. A notable difference is that, according to Ref. [14], the density matrix should be redefined to be non-Hermitian. In the PT -symmetry-broken regime, coherent oscillations in $n(k,t)$ disappear and are turned into a monotonic decay into the steady state, leading to a trivial spin texture.

While exploring (Floquet) topological phases using photonic quantum walks is not new (since we already have Refs. [20-27]), to the best of my knowledge, it is new to apply this setup to simulate quench dynamics, of which the topological aspects have been seriously studied only very recently. In the context of quantum quenches, this should be one of the first experiments (another is Ref. [33] from the same group) that demonstrates a (i) symmetry-protected and (ii) non-unitary dynamic topological phenomenon. Having in mind the growing interest in nonequilibrium, especially non-Hermitian/non-unitary topological phenomena, I recommend the publication in Nature Communication, provided that the following issues are addressed appropriately:

1. In addition to the PT symmetry, the system seems to exhibit a particle-hole symmetry $KUK^{-1} = U$, where K is the complex conjugation. According to Ref. [26] (and also arXiv:1609.09650), the complete topological characterization should be $Z \oplus Z$. I wonder why the authors only focus on one of the integers in $Z \oplus Z$. Is it because that the particle-hole symmetry is irrelevant and the topological number protected by the PT symmetry alone is Z ?

2. As the authors said, what is directly accessible in the experiment is $\rho'(k,t)$ rather than $\rho(k,t)$, which means that $\rho'(k,t)$ is physical while $\rho(k,t)$ is artificial. Why is it more suitable to use the artificial density matrix instead of the physical one for visualizing the skyrmion texture in non-unitary dynamics? What is the reason for "skyrmion structures would remain hidden" if we use the physical density matrix?

3. In ultracold-atom experiments (Refs. [16-18]), we start from a many-particle state with the lower band fulfilled. In contrast, the initial state here is a single-particle state localized at $x=0$ with a specific polarization. In my opinion, such a single-particle state can represent the entire band only if the band is completely trivial, i.e., the Bloch states do not depend on k , which is the case by choosing θ_2 to be $-\pi/2$. Therefore, I wonder whether it is possible (at least in principle) to simulate quench dynamics for more general initial band structures. If this could not be done, I would doubt the powerfulness of photonic quantum walks as a quantum simulator for quench

dynamics.

4. Is it possible to create a situation similar to Fig. 3(d) in Ref. [14] in the present experiment? This regime seems more interesting than those of Fig. 4(b) and Fig. 5(b), which are either too close to the unitary limit or too deep in the PT-symmetry-broken regime.

5. Two minor points regarding the section “Fixed points and dynamical Chern numbers in QW dynamics”:

(1) There are some typos in the expressions of $|\psi_k(t)\rangle$ and $|\chi_k(t)\rangle$ below Eq. (5).

(2) Unlike the PT-symmetric SSH model in Ref. [14], the quasienergies of U^f_k do not take the form $\pm E^f_k$ in general (since d_0 is generally nonzero).

Reviewer #3 (Remarks to the Author):

The paper deals with dynamical effects of topological nature, manifesting in systems with topological phases, where there is a quench between two evolutions (Hamiltonians), belonging to different phases. Here this is studied in the framework of Floquet systems in particular of non-unitary quantum walks.

The topic of dynamical observables that can be considered topological, and manifest after quenches between quantum dynamics featuring topological phases, is promising and is attracting wide interest. Quantum walks are well known systems hosting topological phases, and they are starting to be used also in this context, often referred to as dynamical topological phase transitions [Reports Prog. Phys. 81, 054001 (2018)].

The presence of skyrmions in the spin texture associated with the momentum-time manifold of a 1D topological system is one of these dynamical effects, and has been recently described in a few papers, as described by the authors in the manuscript. In this article, they investigate such phenomenon in the context of quantum walks, in particular of non-unitary quantum walks with parity-time symmetry.

To my knowledge this is the first experiment reporting the presence of skyrmions in QWs, but has to be considered within a series of experiments such as “Wang, K. et al. Simulating dynamic quantum phase transitions in photonic quantum walks. arXiv:1806.10871 1–13 (2018)” (by almost the same authors) and “Xu, X.-Y. et al. Measuring a Dynamical Topological Order Parameter in Quantum Walks. arXiv:1808.03930 1–19 (2018)”, that have studied quenched QWs, when the quench is between two Hamiltonians that feature topological phases.

The paper looks clear in most of its aspects, presented results are original, the derivations are sound and the experiments are well conducted. In order to consider their article for publication in Nature Communications, I would ask the authors to address a series of major and minor remarks, that I am listing below. My main concern is that I feel that presented results represent the experimental verification of a specific phenomenon that was already demonstrated theoretically. Of course, this is an important result by itself and definitively deserves publication, but perhaps it does not meet the criteria for publication in Nature Communication.

Major remarks

Although this is the first experiment showing skyrmions in quenched QWs, there are other experiments investigating the same process in order to detect other topological features, such as

“Wang, K. et al. Simulating dynamic quantum phase transitions in photonic quantum walks. arXiv:1806.10871 1–13 (2018)” (by almost the same authors) and “Xu, X.-Y. et al. Measuring a Dynamical Topological Order Parameter in Quantum Walks. arXiv:1808.03930 1–19 (2018)”. Authors should make clear that their work falls within a broader set of experiments on quenched QWs, and probably discuss connections.

Are all results presented here already predicted theoretically?

The article deals with non-unitary QW, but the authors demonstrate that the presence of skyrmions in the momentum-time space is characterizing the unitary process also. Are there fundamental differences in the appearance of skyrmions in the two different regimes?

Skyrmions are shown in a limited set of configurations. What would happen if the initial Hamiltonian is non-trivial and the final is trivial? And what happens if both are non-trivial, having for instance $\nu = -2$ and $\nu = 2$?

Authors use here a coarser topological classification of their system, that is hosting Floquet topological phases characterized by a couple of invariants (ν_0, ν_π) , as shown for a similar system in [Nat. Phys. 13, 1117–1123 (2017)]. Why aren't they referring to the complete topological classification?

Skyrmions appear only at specific times; what is the mechanism ruling the connection between initial and final Hamiltonians and such “critical” times?

Minor remarks

Details about the experimental procedure that is adopted to detect $\mathbf{n}(\mathbf{k})$ are reported in the Methods and in the Supplementary Material. Perhaps some of these aspects could be moved to the main text to help the reader in understanding with more ease the derivation of the results.

In Figs. 3,4, authors should add the meaning of the color legend adopted for the arrows

I would encourage the authors to provide a clearer explanation of the link between density operators ρ and ρ'

Reply to Report of Reviewer 1

We agree with this summary of our manuscript and appreciate Reviewer 1's endorsement of the novelty, innovation and soundness of our work. The reviewer found our experiment original and interesting, and recommended it for publication in *Nature Communications*. Having followed the reviewer's suggestion and improved the presentation of the paper, we hope the reviewer will be satisfied with the revised manuscript.

Reply to Report of Reviewer 2

We thank Reviewer 2 for her/his careful reading of our manuscript and for recommending our work for publication in *Nature Communications*. Following the reviewer's insightful comments and helpful suggestions, we have significantly revised and improved the manuscript, including conducting a series of new experiments and adding new discussion both in the main text and in Supplemental Materials. We believe that these valuable additions greatly improve the quality of our work, and we hope the reviewer will be satisfied with them. In the following, we reproduce the reviewer's report verbatim and address the reviewer's comments point-by-point.

The authors report an experiment on observing momentum-time skyrmions in parity-symmetric non-unitary photonic quantum walks. This experiment basically demonstrates the theoretical predictions in Refs. [12-14], which unveil the dynamic Chern number and momentum-time skyrmions underlying the nonequilibrium dynamics of one-dimensional free fermions quenched across different symmetry-protected topological phases. Unlike the continuous quench dynamics considered in Refs. [12-14], the authors deal with a unitary U with fine-tuned parameters such that U^6 is (proportional to) the identity. This unitary thus stroboscopically mimics a flat-band quench dynamics.

We agree with the reviewer's summary of our work. The reviewer is right that the current experiment goes beyond the theoretical predictions in Refs. [12-14] by characterizing and experimentally confirming the existence of dynamic skyrmions in discrete-time quench dynamics. Whereas most of our experiments are focused on flat-band quench dynamics for simpler experimental design (see Figs. 3, 4 and 5 in the main text and Figs. S2 and S5 in the revised Supplemental Materials), we have also performed experiments involving more general quench dynamics (Figs. 6(c)(d), S3 and S4 in the revised version), where $(U^f)^6$ is not proportional to identity.

In the presence of non-unitarity, which is controlled by a HWP-PPBS-HWP setup, almost the same skyrmion spin textures $\mathbf{n}(k, t)$ as the unitary case are observed in the \mathcal{PT} -symmetry-unbroken regime. A notable difference is that, according to Ref. [14], the density matrix should be redefined to be non-Hermitian.

In the \mathcal{PT} -symmetry-broken regime, coherent oscillations in $\mathbf{n}(k, t)$ disappear and are turned into a monotonic decay into the steady state, leading to a trivial spin texture.

We agree with the reviewer's summary here. An additional important and observable difference between the unitary and non-unitary dynamics is the deviation of dynamic fixed points in the non-unitary dynamics from high-symmetry points, which leads to deformed skyrmion-lattice structures. This is clearly demonstrated in Figs. 3 and 4.

While exploring (Floquet) topological phases using photonic quantum walks is not new (since we already have Refs. [20-27]), to the best of my knowledge, it is new to apply this setup to simulate quench dynamics, of which the topological aspects have been seriously studied only very recently. In the context of quantum quenches, this should be one of the first experiments (another is Ref. [33] from the same group) that demonstrates a (i) symmetry-protected and (ii) non-unitary dynamic topological phenomenon. Having in mind the growing interest in nonequilibrium, especially non-Hermitian/non-unitary topological phenomena, I recommend the publication in *Nature Communications*, provided that the following issues are addressed appropriately:

We agree with the reviewer that our work is among the first experiments to simulate quench dynamics using discrete-time quantum walks (see also Refs. [29] and [30]). The reviewer is also correct that our experiment is the first observation of momentum-time skyrmions in quantum-walk dynamics (both unitary and non-unitary), which offers an intriguing playground combining previous studies of non-Hermitian topological systems and topological phenomena in quench dynamics. We thank the reviewer for recommending our work for publication and will address the issues raised by the reviewer in details.

1. In addition to the \mathcal{PT} symmetry, the system seems to exhibit a particle-hole symmetry $KUK^{-1} = U$, where K is the complex conjugation. According to Ref. [26] (and also arXiv:1609.09650), the complete topological characterization should be $Z \oplus Z$. I wonder why the authors only focus on one of the integers in $Z \oplus Z$. Is it because that the particle-hole symmetry is irrelevant and the topological number protected by the \mathcal{PT} symmetry alone is Z ?

This is an excellent point, and we thank the reviewer for raising the issue. The reviewer is correct that our system has a particle-hole symmetry, and its complete topological classification should be $Z \oplus Z$, giving rise to a pair of integer topological invariants. According to arXiv:1609.09650 (we have cited this work in the revised version as Ref. [45]) and Ref. [46], these topological invariants can be calculated from Floquet operators in different time frames, which are distinct time-shifted Floquet operators in the periodic sequence of the full discrete-time dynamics. For our system, the other topological invariant is given, in the revised version, by the winding number of U' in Methods and by Eq. (S5) in the Supplemental Materials.

In the previous version, we only considered one of the two topological invariants by implementing U , but, in this revised version, we treat both topological invariants for completeness. Whereas Floquet operators in the two different time frames are inequivalent, and each gives rise to distinct spin textures and momentum-time skyrmions in quench processes, it is sufficient to self-consistently characterize the relation between momentum-time skyrmions and pre- and post-quench winding numbers within a single time frame. In the revised manuscript, we explicitly demonstrate this point by performing a series of new experiments using U' (see Figs. S2 and S3). We show that the emergence of skyrmions under U' is related to pre- and post-quench winding numbers of U' while the system is in the \mathcal{PT} -symmetry-unbroken regime.

In the revised manuscript, we focus on discussion of U in the main text and present new results using U' in the Supplemental Materials. We have also added discussions on this issue in Methods.

2. As the authors said, what is directly accessible in the experiment is $\rho'(k, t)$ rather than $\rho(k, t)$, which means that $\rho'(k, t)$ is physical while $\rho(k, t)$ is artificial. Why is it more suitable to use the artificial density matrix instead of the physical one for visualizing the skyrmion texture in non-unitary dynamics? What is the reason for “skyrmion structures would remain hidden” if we use the physical density matrix?

We thank the reviewer for the excellent question. In fact, both $\rho'(k, t)$ and $\rho(k, t)$ are experimentally accessible. For the detection of $\rho'(k, t)$, we have performed both interference-based measurements and projective measurements to access $|\psi_k(t)\rangle$, as we have detailed in

the manuscript. The detection of $\rho(k, t)$ can be achieved in a similar fashion, but we would need to access both $|\psi_k(t)\rangle$ and $|\chi_k(t)\rangle$, and construct $\rho(k, t)$ by taking the adjoint of $|\chi_k(t)\rangle$. Therefore, compared to the detection of $\rho'(k, t)$, detection of $\rho(k, t)$ requires twice the effort but is definitely experimentally achievable.

More importantly, dynamic skyrmion structures also emerge in the spin texture $\mathbf{n}'(k, t) = \text{Tr}[\rho'_k(t)\boldsymbol{\sigma}]$ for the Pauli vector $\boldsymbol{\sigma} = (\sigma_1, \sigma_2, \sigma_3)$. However, under non-unitary dynamics, right eigenvectors $|\psi_{k,\mu}\rangle$ are not orthogonal to their adjoints, with $\langle\psi_{k,\nu}|\psi_{k,\mu}\rangle \neq \delta_{\nu,\mu}$ under the standard Hilbert-space inner product, and the norms of $|\psi_k(t)\rangle$ and $\mathbf{n}'(k, t)$ are not identity. As a result, $\mathbf{n}'(k, t)$ can only be visualized on a Bloch sphere by enforcing normalization condition in each k -sector. Even then, characterizing the dynamics of $\mathbf{n}'(k, t)$ on the Bloch sphere remains cumbersome, and it is difficult to establish a connection between dynamic skyrmions structures and winding numbers. On the other hand, by introducing the non-Hermitian density matrix $\rho(k, t)$, $\mathbf{n}(k, t) = \text{Tr}[\rho_k(t)\boldsymbol{\tau}]$ is naturally real and of unit norm. We are then able to demonstrate the relation between dynamic skyrmions and topological invariants of Floquet operators in an elegant way using biorthogonal construction (see Methods and Supplemental Materials). We also note that momentum-time spin textures $\mathbf{n}(k, t)$ and $\mathbf{n}'(k, t)$ differ, but the conditions for the emergence of dynamic skyrmions are the same, as explicitly shown in Fig. S5 using experimental data.

Thus, we agree with the reviewer that it is confusing to claim “skyrmions structures would remain hidden” when we choose $\rho'(k, t)$. In the revised version, we modify the discussion and added both theoretical and experimental characterization of the relation between the two density matrices (see revised Supplemental Materials).

3. In ultracold-atom experiments (Refs. [16-18]), we start from a many-particle state with the lower band fulfilled. In contrast, the initial state here is a single-particle state localized at $x = 0$ with a specific polarization. In my opinion, such a single-particle state can represent the entire band only if the band is completely trivial, i.e., the Bloch states do not depend on k , which is the case by choosing θ_2^i to be $-\pi/2$. Therefore, I wonder whether it is possible (at least in principle) to simulate quench dynamics for more general initial band structures. If this could not be done, I would doubt the powerfulness of photonic quantum walks as a quantum simulator for quench dynamics.

This is an excellent comment. The reviewer is right that our initial state, being a localized single-particle state, is necessarily topologically trivial. However, it is also possible to initialize the state which represents topologically non-trivial band structures. This requires preparing a non-local or quasi-local single-particle state, which can be achieved by subjecting an initially localized single-photon state to appropriately-designed gate operations. In the revised manuscript, we have performed new experiments to explicitly demonstrate this generality by simulating quench dynamics starting from a topologically non-trivial state. As shown in Fig. 5 of the main text, we simulate quench dynamics between Floquet topological phases with $\nu^i = -2$ and $\nu^f = 2$, where rich skyrmion-lattice structure emerges. The initial state is given by $|\psi_-^i\rangle = (-0.1111 - 0.6983i) |0\rangle \otimes |+\rangle - \frac{1}{\sqrt{2}} |-2\rangle \otimes |-\rangle$, which is quasi-local and easily prepared by performing gate operations prior to the quantum-walk dynamics (see the revised Methods for details). In principle, more general initial band structures can be prepared in a similar fashion, though careful designs of gate operations are required.

Further, the strength of photonic quantum walks as quantum simulators of quench dynamics also lies in their versatile controls which are difficult to implement in other synthetic systems such as cold atoms. Straightforward examples include quench dynamics of mixed states or entangled two-photon states, which allows for exploring the interplay of coherence or entanglement with dynamic topological phenomena. Another example is non-unitary quench dynamics which is exactly what we have achieved here. We therefore believe that photonic quantum walks represent a powerful platform for the simulation of quench dynamics.

Is it possible to create a situation similar to Fig. 3(d) in Ref. [14] in the present experiment? This regime seems more interesting than those of Fig. 4(b) and Fig. 5(b), which are either too close to the unitary limit or too deep in the \mathcal{PT} -symmetry-broken regime.

It is possible to create a situation similar to Fig. 3(d) in Ref. [14], which we show in Figs. 6(c)(d) of the revised manuscript. In Figs. 6(a)(b), we quench the system into a regime where \mathcal{PT} symmetry is completely broken and U^f has a completely imaginary quasienergy spectrum. In contrast, in the new experiment shown in Figs. 6(c)(d), we quench the system into the \mathcal{PT} -symmetry-partially-broken regime where quasienergies of U^f are imaginary within a certain range of momenta and are real otherwise. The results are similar to Fig. 3(d)

in Ref. [14], though a richer skyrmion structure is present in our experiment due to larger winding numbers.

Two minor points regarding the section “Fixed points and dynamical Chern numbers in QW dynamics”:

- (1) There are some typos in the expressions of $|\psi_k(t)\rangle$ and $|\chi_k(t)\rangle$ below Eq. (5).
- (2) Unlike the \mathcal{PT} -symmetric SSH model in Ref. [14], the quasienergies of U_k^f do not take the form $\pm E_k^f$ in general (since d_0 is generally nonzero).

We thank the reviewer for pointing out these issues. We have corrected the typo following Eq. (5). Regarding quasienergies of U_k^f , we agree with the reviewer that d_0 is generally nonzero. However, since eigenvalues of U_k^f satisfy $\lambda_{k,+}\lambda_{k,-} = 1$ even with nonzero d_0 [expressions of $\lambda_{k,\pm}$ are given below Eq. (3)], they can still be written in the form of $\pm E_k^f$.

Reply to Report of Reviewer 3

We thank Reviewer 3 for her/his careful reading of our manuscript and for the helpful comments and suggestions. Whereas the reviewer finds our work original and important, her/his main concern is whether our results “represent the experimental verification of a specific phenomenon that was already demonstrated theoretically.” We welcome the opportunity to clarify, in our revised manuscript, that both our theoretical predictions and experiment schemes are of new and innovative. With these latest improvements, we believe that our work is original and appropriate for publication in *Nature Communications*. In the following, we address the reviewer’s comments point-by-point.

The paper deals with dynamical effects of topological nature, manifesting in systems with topological phases, where there is a quench between two evolutions (Hamiltonians), belonging to different phases. Here this is studied in the framework of Floquet systems in particular of non-unitary quantum walks.

We agree with the reviewer’s general summary. More specifically, our experiment reports the first observation of dynamic skyrmions in discrete-time quantum dynamics.

The topic of dynamical observables that can be considered topological, and manifest after quenches between quantum dynamics featuring topological phases, is promising and is attracting wide interest. Quantum walks are well known systems hosting topological phases, and they are starting to be used also in this context, often referred to as dynamical topological phase transitions [Reports Prog. Phys. 81, 054001 (2018)].

We agree with the reviewer that the topic of dynamic topological phenomena in quench dynamics has attracted wide interest recently. In the revised manuscript, we added further discussions on the context of our experiment, where we cited [Reports Prog. Phys. 81, 054001 (2018)].

The presence of skyrmions in the spin texture associated with the momentum-time manifold of a 1D topological system is one of these dynamical effects, and has been recently described in a few papers, as described by the authors in the manuscript. In this article, they investigate such phenomenon in the context

of quantum walks, in particular of non-unitary quantum walks with parity-time symmetry.

We agree with the summary of our work. We emphasize that dynamic skyrmions reported in this work are distinct topological constructions from previously reported dynamic quantum phase transitions and dynamical topological order parameters in quantum walks.

To my knowledge this is the first experiment reporting the presence of skyrmions in QWs, but has to be considered within a series of experiments such as “Wang, K. et al. Simulating dynamic quantum phase transitions in photonic quantum walks. arXiv:1806.10871 113 (2018)” (by almost the same authors) and “Xu, X.-Y. et al. Measuring a Dynamical Topological Order Parameter in Quantum Walks. arXiv:1808.03930 119 (2018)”, that have studied quenched QWs, when the quench is between two Hamiltonians that feature topological phases.

We agree with the referee that our experiment is the first observation of dynamic skyrmions in non-unitary quench processes. Importantly, our experiment differs fundamentally from previous experimental studies of quenched quantum walks, both in theoretical description and in experimental scheme. Thus, our work is not part of a series of existing experiments (see our reply to the reviewer’s detailed remarks below).

The paper looks clear in most of its aspects, presented results are original, the derivations are sound and the experiments are well conducted. In order to consider their article for publication in *Nature Communications*, I would ask the authors to address a series of major and minor remarks, that I am listing below. My main concern is that I feel that presented results represent the experimental verification of a specific phenomenon that was already demonstrated theoretically. Of course, this is an important result by itself and definitely deserves publication, but perhaps it does not meet the criteria for publication in *Nature Communications*.

We thank the reviewer for finding our experiment original and important. Our experiment represents the first time that a two-dimensional dynamic topological structure has been observed using quantum-walk dynamics, which is achieved by employing new experimental

schemes. Since dynamic skyrmions were not yet predicted to exist in discrete-time Floquet dynamics, both our theoretical prediction and experimental implementation are new and innovative. Furthermore, compared to other synthetic systems exhibiting rich dynamic topological structures in higher dimensions (see Refs. [16-19]), prior studies of dynamic topological structures in quantum walks have been limited to one dimension. Our work thus represents an important advance for quantum simulations using quantum walks.

Given this novelty, we believe that our experiment is original and appropriate for publication in *Nature Communications*. In the following, we address the reviewer’s remarks in detail.

Although this is the first experiment showing skyrmions in quenched QWs, there are other experiments investigating the same process in order to detect other topological features, such as “Wang, K. et al. Simulating dynamic quantum phase transitions in photonic quantum walks. arXiv:1806.10871 113 (2018)” (by almost the same authors) and “Xu, X.-Y. et al. Measuring a Dynamical Topological Order Parameter in Quantum Walks. arXiv:1808.03930 119 (2018)”. Authors should make clear that their work falls within a broader set of experiments on quenched QWs, and probably discuss connections.

Our work reports the first observation of two-dimensional emergent topological phenomena in quantum-walk dynamics, which differs fundamentally from previous experimental studies of one-dimensional dynamic topological structures in quenched quantum walks. It is therefore not part of a series, but novel in both theoretical description and experimental implementation.

First, dynamic skyrmion structures studied here are distinct in construction from dynamic topological order parameters characterizing dynamic topological phase transitions explored in arXiv:1806.10871 and arXiv:1808.03930, which are cited as Refs. [29] and [30] in the revised version. Dynamic topological order parameters are winding numbers characterizing an $S^1 - S^1$ mapping defined through the Pancharatnam geometric phase over a one-dimensional Brillouin zone. In contrast, dynamic skyrmions structures are protected by dynamic Chern numbers, which characterize $S^2 - S^2$ mappings defined through the density matrix over two-dimensional momentum-time submanifolds. Our work is the first time that a two-dimensional dynamic topological structure has been theoretically predicted and exper-

imentally observed in quantum-walk dynamics, which greatly extends the scope of quantum simulation using quantum walks.

Second, we devised new experimental schemes that are crucial in revealing dynamic skyrmions. Specifically, we construct time-dependent momentum-space density matrices of spatially non-localized states by performing a combination of interference-based measurements and projective measurements in position space. This approach is in sharp contrast to experimental schemes in Refs. [29] and [30], where the observation of dynamic topological order parameters does not require a complete knowledge of density matrices and it is sufficient to perform interference-based measurements to probe the inner products between initial and final states.

In the revised manuscript, we have followed the reviewer's suggestion and added discussion (both in the Introduction and in the Discussion sections) on the connection and the fundamental differences between our experiment and these two previous experiments. We have also modified the abstract to highlight key novelties of our work.

Are all results presented here already predicted theoretically?

Results in our paper were not previously predicted theoretically. Prior to our experiment, dynamic skyrmions were not yet predicted to exist in discrete-time Floquet dynamics such as quantum walks. In previous works, dynamic skyrmions have only been investigated for quenches of one-dimensional lattice models such as the Su-Schrieffer-Heeger model or the Kitaev chain (see Refs. [12-14]). Whereas these theoretical works are related to our observation, dynamic skyrmions in discrete-time Floquet dynamics are different, both in theoretical characterization and in appearance, from those in lattice models. For example, the presence of two distinct topological invariants in Floquet dynamics requires the introduction of two time frames, each with its own Floquet operator and skyrmion structures. Furthermore, the existence of larger topological invariants in quantum-walk dynamics gives rise to richer skyrmion-lattice structures compared to those in previously studied lattice models.

# The article deals with non-unitary QW, but the authors demonstrate that the presence of skyrmions in the momentum-time space is characterizing the unitary process also.

We agree with the reviewer that momentum-time skyrmions can exist in both unitary and non-unitary quench processes.

Are there fundamental differences in the appearance of skyrmions in the two different regimes?

There are two fundamental differences between the appearance of skyrmions in unitary and non-unitary quench processes.

First, whereas skyrmions appear in unitary quench processes as long as the pre- and post-quench winding numbers are different, skyrmions in non-unitary quench processes generically do not exist in \mathcal{PT} -symmetry-spontaneously-broken regimes even if the pre- and post-quench winding numbers are different. This is illustrated experimentally in Fig. 6 of the main text.

Second, locations of topologically protected fixed points are different in the two cases, which affect the overall appearance of the skyrmion-lattice structure in momentum-time space. In the unitary case, fixed points are located at high-symmetry points in momentum space, whereas in the non-unitary case they are shifted away from those high-symmetry points. Since fixed points divide momentum-time space into multiple submanifolds on which dynamic skyrmions appear, the skyrmion-lattice structures in non-unitary quenches are no longer evenly distributed, but deformed along the momentum manifold. This is shown in Fig. 4 of the main text.

In the revised manuscript, we have added new experiments [see Figs. 6(c)(d)] demonstrating more quench processes into the \mathcal{PT} -symmetry-broken regime, where the contrast with unitary dynamics is clearer.

Skyrmions are shown in a limited set of configurations. What would happen if the initial Hamiltonian is non-trivial and the final is trivial?

Theoretically, skyrmion structures should emerge when topological invariants of the initial and final Floquet operators are different, and when both Floquet operators are in the \mathcal{PT} -symmetry-unbroken regime. In this case, the total number of topologically protected fixed points with $c_+ = 0$ or $c_- = 0$ is $|\nu^f - \nu^i|$ each (see Methods). Dynamic Chern numbers are finite on the submanifold between two adjacent fixed points of different kinds (plus the S^1 temporal manifold), which protect dynamic skyrmions on the corresponding momentum-time submanifolds.

Therefore, if the initial Floquet operator is topologically non-trivial and the final is trivial, fixed points and dynamic skyrmions should persist, provided both Floquet operators are in the \mathcal{PT} -symmetry-unbroken regime.

And what happens if both are non-trivial, having for instance $\nu = -2$ and $\nu = 2$?

If both the initial and final Floquet operators are topological non-trivial, skyrmions can only exist when their topological invariants are different. But a larger winding-number difference between the initial and final Floquet operators gives rise to more topologically protected fixed points and hence a richer skyrmion-lattice structure.

For instance, when $\nu^i = -2$ and $\nu^f = 2$, there would be at least 4 pairs of fixed points of different kinds. Momentum-time skyrmions then appear on the momentum-time manifolds between adjacent fixed points, giving rise to a rich skyrmions-lattice structure. This is experimentally confirmed in the revised manuscript, where we have performed new experiments to explicitly demonstrate this point (see Fig. 5 of the main text and the corresponding discussion). We thank the reviewer for this interesting suggestion.

Authors use here a coarser topological classification of their system, that is hosting Floquet topological phases characterized by a couple of invariants (ν_0, ν_π) , as shown for a similar system in [Nat. Phys. 13, 1117-1123 (2017)]. Why aren't they referring to the complete topological classification?

This is an excellent point and we thank the reviewer for raising the issue. The complete topological classification of our system should be $Z \oplus Z$, giving rise to a pair of integer topological invariants. According to Refs. [45] and [46] in the revised manuscript, these topological invariants can be calculated from Floquet operators in different time frames, which are different time-shifted Floquet operators in the periodic sequence of the full discrete-time dynamics. For our system, the other topological invariant is given, in the revised version, by the winding number of U' in Methods and by Eq. (S5) in the Supplemental Materials.

In the previous version, we only considered one of the two topological invariants by implementing U , but, in this revised version, we treat both topological invariants for completeness. Whereas Floquet operators in the two different time frames are inequivalent and each give rise to distinct spin textures and momentum-time skyrmions in quench processes, we find that it is sufficient to self-consistently characterize the relation between momentum-time skyrmions and pre- and post-quench winding numbers within a single time frame. In the revised manuscript, we explicitly demonstrate this point by performing a series of new experiments using U' (see Figs. S2 and S3). We show that the emergence of skyrmions under

U' is related to pre- and post-quench winding numbers of U' while the system is in the \mathcal{PT} -symmetry-unbroken regime.

As the main conclusions regarding dynamic skyrmions are qualitatively similar in either time frame, we opt to focus on the discussion of U in the main text and present new results using U' in the Supplemental Materials. We have also added discussions on the issue in Methods.

Skyrmions appear only at specific times;

Dynamic skyrmions are dictated by dynamic Chern numbers defined on the two-dimensional momentum-time submanifold between two adjacent fixed points. Therefore by definition, dynamic skyrmions are non-local structures in momentum-time space, and cannot be regarded as occurring at specific times.

what is the mechanism ruling the connection between initial and final Hamiltonians and such “critical” times?

The mechanism behind Floquet operators and dynamic skyrmions is the following. If both initial and final Floquet operators are in the \mathcal{PT} -symmetry-unbroken regime, the absolute value of their winding-number difference gives the number of pairs of fixed points. Dynamic Chern numbers are finite on the submanifold between two adjacent fixed points of different kinds (plus the S^1 temporal manifold), which protect dynamic skyrmion structures in the corresponding submanifold. We emphasize that these skyrmions do not occur at specific critical times.

Details about the experimental procedure that is adopted to detect $\mathbf{n}(k)$ are reported in the Methods and in the Supplementary Material. Perhaps some of these aspects could be moved to the main text to help the reader in understanding with more ease the derivation of the results.

We have followed the reviewer’s suggestion and moved part of the discussion on experimental procedure to the main text, at the end of the section “Fixed points and emergent skyrmions”.

In Figs. 3 and 4, authors should add the meaning of the color legend adopted for the arrows.

We thank the reviewer for the suggestion. We have modified these figures accordingly.

I would encourage the authors to provide a clearer explanation of the link between density operators ρ and ρ' .

We have followed the reviewer's suggestion and added more discussions both in the main text and in Supplemental Materials to clarify the relation between $\rho(k, t)$ and $\rho'(k, t)$.

Both $\rho(k, t)$ and $\rho'(k, t)$ can be accessed experimentally, and dynamic skyrmions can be visualized starting with either density matrix (albeit with completely different treatment). The adoption of the non-Hermitian $\rho(k, t)$ here is because by invoking the biorthogonal formalism and choosing $\rho(k, t)$, it is much easier to demonstrate the existence and locations of fixed points and relate dynamic skyrmions to pre- and post-quench topological invariants.

In the revised Supplemental Materials, we explicitly demonstrate the relation between the two density matrices.

List of changes

1. We have conducted a series of new experiments to demonstrate dynamic skyrmions in the alternative time frame given by U' . We have added new figures and discussion in the Supplemental Materials on the subject.
2. We have conducted new experiments to reflect dynamics in the \mathcal{PT} -symmetry-broken region. Corresponding new figures are added to Fig. 6 of the main text.
3. We have conducted new experiments simulating quench dynamics between topological non-trivial Floquet topological phases. Corresponding new figures are added as Fig. 5 of the revised manuscript. We have also added discussion in the main text on the figure.
4. In the Methods section, we have added discussion on the initial-state preparation for Fig. 5.
5. We have added discussion on the difference and connection between ρ and ρ' throughout the revised manuscript, including a new figure and a new section in the Supplemental Materials as well as discussion in the main text and the Methods section.
6. We have added discussion in the Introduction and the Discussion sections to compare and differentiate our experiment from previous experiments on the observation of dynamic quantum phase transitions.
7. We have moved part of the discussion on experimental procedure to the main text, as suggested by Reviewer 3.
8. Following the suggestion of Reviewer 3, we have re-plotted all figures with spin textures, so that the color of spin vectors represents the value of $\mathbf{n}_3(k, t)$.
9. We have added more references to accompany the added discussions. We have also updated various references.
10. We have corrected various typos as pointed out by the reviewers.
11. We have improved the presentation of the manuscript so that it is less technical and more readable.

REVIEWERS' COMMENTS:

Reviewer #2 (Remarks to the Author):

The authors have made a great effort to address my major criticisms in the previous report. In particular, the authors have performed (extensively) additional experiments to confirm the consistency between the emergence of momentum-time skyrmions and the change of winding number in an alternative time frame (Figs. S2 and S3), where the winding number corresponds to the other integer in the full $Z \oplus Z$ classification. The more interesting regime with partial PT -symmetry breaking has also been demonstrated (Figs. 6(c) and (d)). I am now convinced that both ρ and ρ' reflect the topology of the quench dynamics and are both experimentally accessible, while using ρ is more convenient since the n vector is naturally normalized. Also, now I understand that the photonic quantum walk is capable of simulating the quench dynamics starting from nontrivial bands, as long as we choose the initial state to be a Wannier function. There is no doubt that the quality of the revised manuscript is greatly improved. I am quite satisfied with the present version and recommend its publication in Nature Communication.

Reviewer #3 (Remarks to the Author):

The manuscript contains many improvements with respect to the original version. The authors replied in details to all comments by me and the other reviewers. I think the paper is now suitable for publication in Nature Communications.

This said, there is still one aspect that does not convince me totally. All the paper is focused on non-unitary quantum walks with PT symmetry, and in this context the authors investigate the emergence of dynamic skyrmions in the (k,t) manifold. As the authors explain, this was known in generic quench dynamics, but was not studied in Floquet systems such as quantum walks, and I agree with this. The authors show also one example of the same feature emerging in unitary QWs, see fig. 3a and 4a. In my opinion, these 2-d topological structures (the skyrmions) look very similar in quenches between unitary evolutions or non-unitary ones. Of course, I understand that no structures are observed when the QW is in the PT -broken phase. But when we are in the unbroken regime, I can't see fundamental differences between the two configurations. If the authors agree on this perspective, I think that they could make this point much clearer in the paper, this would be extremely helpful for the reader. This is also motivated by the fact that such skyrmions were not studied so far in unitary QWs as well, so they look as a novelty by themselves.

We thank Reviewer 2 and Reviewer 3 again for their helpful comments and for recommending our work for publication in Nature Communications. In the following, the reviewers' reports are reproduced verbatim in blue font, and our point-by-point responses are in black.

Reply to Review 2:

The authors have made a great effort to address my major criticisms in the previous report. In particular, the authors have performed (extensively) additional experiments to confirm the consistency between the emergence of momentum-time skyrmions and the change of winding number in an alternative time frame (Figs. S2 and S3), where the winding number corresponds to the other integer in the full $Z \oplus Z$ classification. The more interesting regime with partial PT -symmetry breaking has also been demonstrated (Figs. 6(c) and (d)). I am now convinced that both ρ and ρ' reflect the topology of the quench dynamics and are both experimentally accessible, while using ρ is more convenient since the \mathbf{n} vector is naturally normalized. Also, now I understand that the photonic quantum walk is capable of simulating the quench dynamics starting from nontrivial bands, as long as we choose the initial state to be a Wannier function. There is no doubt that the quality of the revised manuscript is greatly improved. I am quite satisfied with the present version and recommend its publication in Nature Communication.

We thank Reviewer 2 for finding our manuscript greatly improved and for recommending the work for publication in Nature Communications.

Reply to Reviewer 3:

The manuscript contains many improvements with respect to the original version. The authors replied in details to all comments by me and the other reviewers. I think the paper is now suitable for publication in Nature Communications.

We thank Reviewer 3 for recommending our work for publication in Nature Communications.

This said, there is still one aspect that does not convince me totally. All the paper is focused on non-unitary quantum walks with PT symmetry, and in this context the authors investigate the emergence of dynamic skyrmions in the (k,t) manifold. As the authors explain, this was known in generic quench dynamics, but was not studied in Floquet systems such as quantum walks, and I agree with this. The authors show also one example of the same feature emerging in unitary QWs, see fig. 3a and 4a. In my opinion, these 2-d topological structures (the skyrmions) look very similar in quenches between unitary evolutions or non-unitary ones. Of course, I understand that no structures are observed when the QW is in the PT -broken phase. But when we are in the unbroken regime, I can't see fundamental differences between the two configurations. If the authors agree on this perspective, I think that they could make this point much clearer in the paper, this would be extremely helpful for the reader. This is also motivated by the fact that such skyrmions were not studied so far in unitary QWs as well, so they look as a novelty by themselves.

We agree with the reviewer that an explicit comparison between skyrmions structures in the unitary and PT-symmetric non-unitary quenches should be very helpful for readers. In the revised manuscript, we have added the following sentence to the discussion of Fig. 4 (on page 5) :

Comparing Figs. 4(a) and (b), we see that dynamic skyrmion structures in the unitary and the PT-symmetric non-unitary quench processes are qualitatively similar; albeit, in the non-unitary case, skyrmion structures are slightly deformed due to the shift of fixed points. However, as we show later, skyrmion structures are generically absent when the system is quenched into the PT-symmetry-broken regime.